# A Gain-of-Function Mutation on *BCKDK* Gene and Its Possible Pathogenic Role in Branched-Chain Amino Acid Metabolism

**DOI:** 10.3390/genes13020233

**Published:** 2022-01-26

**Authors:** Alice Maguolo, Giulia Rodella, Alejandro Giorgetti, Marion Nicolodi, Rui Ribeiro, Alice Dianin, Gaetano Cantalupo, Irene Monge, Sarah Carcereri, Margherita Lucia De Bernardi, Massimo Delledonne, Andrea Pasini, Natascia Campostrini, Florina Ion Popa, Giorgio Piacentini, Francesca Teofoli, Monica Vincenzi, Marta Camilot, Andrea Bordugo

**Affiliations:** 1Department of Mother and Child, University of Verona, I-37126 Verona, Italy; alice.maguolo@univr.it (A.M.); giulia.rodella@univr.it (G.R.); giorgio.piacentini@univr.it (G.P.); francesca.teofoli@univr.it (F.T.); marta.camilot@univr.it (M.C.); 2Inherited Metabolic Diseases Unit and Regional Centre for Newborn Screening, Diagnosis and Treatment of Inherited Metabolic Diseases and Congenital Endocrine Diseases, Azienda Ospedaliera Universitaria Integrata, I-37126 Verona, Italy; alice.dianin@aovr.veneto.it (A.D.); irene.monge@aismme.org (I.M.); carcereris@yahoo.it (S.C.); 3Department of Biotechnology, University of Verona, Strada Le Grazie 15, I-37134 Verona, Italy; alejandro.giorgetti@univr.it (A.G.); marion.nicolodi@studenti.univr.it (M.N.); massimo.delledonne@univr.it (M.D.); 4Institute of Neuroscience and Medicine INM-9, Institute for Advanced Simulations IAS-5, Forschungszentrum Jülich, D-52425 Jülich, Germany; r.ribeiro@fz-juelich.de; 5Pediatric Clinic AOUI of Verona, Verona, I-37126 Italy; margheritaluciadebernardi@gmail.com; 6Child Neuropsychiatry Unit, Azienda Ospedaliera Universitaria Integrata, I-37126 Verona, Italy; gaetano.cantalupo@univr.it; 7Department of Molecular Medicine and Biotecnology, AOU Federico II, I-80131 Napoli, Italy; 8Department of Pediatrics, The Regional Center for Neonatal Screening, Diagnosis and Treatment of Inherited Congenital Metabolic and Endocrinological Diseases, AOUI, I-37134 Verona, Italy; andrea.pasini@aovr.veneto.it (A.P.); natascia.campostrini@aovr.veneto.it (N.C.); florina.ionpopa@aovr.veneto.it (F.I.P.); monica.vincenzi@aovr.veneto.it (M.V.)

**Keywords:** maple syrup urine disease, leucinosis, branched-chain amino acid metabolism, branched-chain ketoacid dehydrogenase kinase, newborn screening, genetic analysis, molecular dynamics simulations, whole-exome sequencing

## Abstract

BCKDK is an important key regulator of branched-chain ketoacid dehydrogenase complex activity by phosphorylating and so inactivating branched-chain ketoacid dehydrogenases, the rate-limiting enzyme of the branched-chain amino acid metabolism. We identified, by whole exome-sequencing analysis, the p.His162Gln variant of the *BCKDK* gene in a neonate, picked up by newborn screening, with a biochemical phenotype of a mild form of maple syrup urine disease (MSUD). The same biochemical and genetic picture was present in the father. Computational analysis of the mutation was performed to better understand its role. Extensive atomistic molecular dynamics simulations showed that the described mutation leads to a conformational change of the BCKDK protein, which reduces the effect of inhibitory binding bound to the protein itself, resulting in its increased activity with subsequent inactivation of BCKDC and increased plasmatic branched-chain amino acid levels. Our study describes the first evidence of the involvement of the *BCKDK* gene in a mild form of MSUD. Although further data are needed to elucidate the clinical relevance of the phenotype caused by this variant, awareness of this regulatory activation of BCKDK is very important, especially in newborn screening data interpretation.

## 1. Introduction

Valine, leucine, and isoleucine, the three branched-chain amino acids (BCAAs), are among the nine essential amino acids. Most dietary proteins consist of about 20–25% BCAAs [1]. BCAAs are hydrophobic amino acids, and they play crucial roles in determining the structures of globular proteins, as well as the interaction of the transmembrane domains of membranous proteins with phospholipid bilayers [2]. Their catabolic disposal occurs largely in the skeletal muscle, and their circulating concentrations can influence the brain uptake of precursor amino acids for neurotransmitter synthesis, especially glutamate [3].

The first step in BCAA catabolism involves the conversion of leucine, isoleucine, and valine into their corresponding branched-chain α ketoacids (BCKAs) by branched-chain aminotransferase (BCAT) located within the mitochondria [3]. The second step consists in the oxidative decarboxylation of BCKAs by the mitochondrial branched-chain α-ketoacid dehydrogenase complex (BCKDC), as demonstrated by Reed and colleagues [4], which represents the thermodynamically irreversible step of BCAA catabolism.

BCKDC is a macromolecular machine that consists in three catalytic components: branch-chain α-ketoacid decarboxylase (E1), existing as an alpha2/beta2 heterotetramer, encoded by the *BCKDHA* and *BCKDHB* genes, respectively; dihydrolipoyl acyltransferase (E2) encoded by the *DBT* gene; and dihydrolipoamide dehydrogenase (E3), encoded by the *DLD* gene [5]. The activity of BCKDC is regulated by both covalent and allosteric mechanisms. It can be allosterically inhibited by NADH and the CoA esters originating from the oxidative decarboxylation of BCKAs, and covalently regulated by two regulatory enzymes, the BCKDC kinase (BCKDK) and the BCKDC phosphatase (BCKDP) [6]. The former inhibits the E1α subunit of the BCKDH protein by phosphorylation at codon Ser292, and the latter (BCKDP) activates it upon dephosphorylation at the same residue [2].

The genes encoding the various catalytic subunits/components (E1α, E1β, E2, E3, kinase, and phosphatase) have been mapped to chromosome loci: 19q13.1–13.2, 6q14, 1p31, 7q31–32, 16p11.2, and 4q22.1, respectively.

A metabolic failure in the oxidative decarboxylation of BCAAs caused by mutations in any of the BCKDC components (*BCKDHA*, *BCKDHB*, *DLD*, or *DBT* genes) results in maple syrup urine disease (MSUD), or branched-chain ketoaciduria (OMIM: 248600) [4,7], with elevations of BCAAs in plasma, α-ketoacids in urine, and alloisoleucine in plasma, the pathognomonic marker of the disease.

The name MSUD originates from a discovery by John Menkes and colleagues in 1954 of four siblings who died of progressive neurological degeneration with severe and early-onset ketoacidosis in the first week of life, and who had a strong maple syrup odor in their urine [7,8].

MSUD is inherited in an autosomal recessive pattern, and despite the absence of a defined genotype–phenotype correlation, on the bases of age at onset, severity of symptoms, response to thiamine supplementation, biochemical findings, and the gene locus ultimately affected [9], there are presently five known clinical phenotypes: classic, intermediate, intermittent, thiamin-responsive, and dihydrolipoamide dehydrogenase (E3)-deficient form.

The age at onset and the severity of symptoms vary according to the residual BCKDC activity. The classic form is the most severe phenotype and occurs in the neonatal period with developmental delay, failure to thrive, feeding difficulties, and maple syrup odor in the urine. It can lead to irreversible encephalopathy, metabolic decompensation, and life-threating complications and is characterized by a 0–2% residual BCKDC activity. Intermediate and intermittent types have milder clinical presentations with a higher residual BCKDC activity [10,11,12]. The intermittent-type and thiamine-responsive forms have a late onset and an episodic presentation during catabolic states.

MSUD occurs in approximately 1:200,000 births [13]. Clinical outcomes are generally good in patients where treatment is initiated early. MSUD is now included in expanded newborn screening (NBS) programs, and elevated plasma BCAAs levels are effectively detected by automated liquid chromatography–tandem mass spectrometry (LC–MS/MS) on dried blood spot (DBS), allowing early diagnosis, timely therapeutic intervention, and a dramatic improvement in the prognosis of this condition [10]. The MSUD-suggestive amino acidic profile in DBS is subsequently confirmed by quantitative plasma aminoacid analysis, enzyme essays of BCKDC activity in cultured fibroblasts, and genetic analysis [14]. 

The therapy for MSUD consists in a dietary restriction of BCAAs using a BCAA-free amino acid mixture and valine and isoleucine supplementation (as the contents of these tend to be lower than leucine in medical foods), thiamine (vitamin B1) supplementation for thiamine-responsive phenotype patients, and liver transplantation, which is curative in some cases [3,11].

## 2. Methods

### 2.1. Clinical Data

We report the case of a term male baby, firstborn of nonconsanguineous Caucasian parents, with a birth weight of 3500 g, a cranial circumference of 34.6 cm, and a regular adaptation to extrauterine life. Expanded NBS by LC–MS/MS performed at 37 h after birth showed DBS BCAAs in the normal range (Val 214 µM, reference range (rr) < 250 µM, [Leu+Ile+AlloIle] 225 µM, rr < 250 µM), but with slightly elevated ratios: Val/Phe 4.21 (rr < 3), [Leu+Ile+AlloIle]/Phe 4.4 (rr < 3.5), and especially [Leu+Ile+AlloIle]/Ala 1.47 (rr < 0.7). The same DBS was then subjected to a second-tier test, aimed at the detection of Alloile, which resulted in 8.2 µM (rr < 2 µM). Its elevation represents a pathognomonic marker for MSUD diagnosis. The following analysis of the plasma patient displayed slightly altered plasma BCAAs (Table 1), uninformative urinary organic acids (see Appendix A), with a consequent suspected diagnosis of mild form of MSUD.

The baby was fed with a controlled amount of normal infant formula and supple-mented with BCAA‐free formula and an oral supplementation of Valine, Isoleucine and Thiamine. Dietary changes were made during the subsequent years using a leucine ex-change system for foods and targeting the diet according to plasma amino acids and clin-ical parameters. The parents were instructed on an emergency diet without leucine, with BCAA-free protein substitute and providing energy requirements, to use just in case of in-tercurrent diseases.The parents and relatives were investigated. Whereas the mother and paternal grandmother’s values were normal, the father and paternal grandfather’s plas-ma BCAAs and Alloile were slightly elevated (Table 2). They are healthy, and so far, no metabolic crises have been reported for both. Anyway, the paternal grandfather had mild-er basal biochemical alterations that were not confirmed at the protein load test, whereas only in the father was a biochemical pattern similar to the patient confirmed (Table 3).

### 2.2. Genetic Analysis

Genomic DNA was extracted from the subject’s peripheral venous blood on EDTA by means of the QIAamp DNA Blood Mini Kit (Qiagen S.p.A, Milan, Italy), following the manufacturer’s instructions. All exons and part of the flanking intron regions of the *BCKDHA* (NM 000709.3), *BCKDHB* (NM_183050.2), *DBT* (NM 001918.3), *DLD* (NM 000108.4), *BCAT2* (NM 001190.4), and *PPM1K* (NM 152542.5) genes were amplified by polymerase chain reactions and sequenced for molecular analysis (primers available upon request) at the Molecular Human Genetic Institute of Monza, showing no pathogenetic variants.

Subsequently, whole-exome sequencing (WES) analysis with next-generation sequencing technology (NGS) was performed (Genomic Lab, University of Verona), confirming no mutations in the patient’s *BCKDHA*, *BCKDHB*, and *DBT* genes, whereas the analysis focused on the regulatory genes identified the heterozygous NM_005881.3:c.486C>A, NP_005872.2:p.His162Gln variant in the *BCKDK* gene (NC_000016.9:g.31121588C>A). The presence of the heterozygous His162Gln variant was confirmed by Sanger sequencing in the newborn and in his father, but not in the *BCKDK* gene of the paternal grandfather (Genetic Lab, LURM, University of Verona).

Several databases have been used to verify whether the variant was already reported in the literature and in which population frequency (ExAC Variant Frequencies 1.0, 1000 Genomes Project, NHLBI ESP 6500, gnomAD Genomes Variant Frequencies 2.1.1, gnomAD Exomes Variant Frequencies 2.1.1, HGMD Professional v2021.1, ClinVar 2021-08-05, dbSNSP 154 v2). To assess the effect of amino acid substitution on the protein, in silico prediction analyses were performed using the following tools: SIFT, Polyphen2 HVAR, MutationTaster, MutationAssessor, FATHMM Pred^©^, FATHMM MKL Coding Pred^©^, BayesDel, addAF, DANN, DEOGEN2, EIGEN, LIST-S2, M-CAP, MVP, and PrimateAI.

### 2.3. Computational Methods

#### 2.3.1. Structure Preparation

For the analysis, we used the rat BCKDK (rBCKDK) crystal structure due to the 95.6% sequence identity with the human sequence (see Appendix A). The starting BCKDK structures (Appendix A) cocrystallized with (2S)-chloro-4-methylpentanoic acid and with the natural inhibitor KIC, respectively, were obtained from PDB (Code 3TZ0: residue 68–404 and Code 4H7Q: residue 74–405) [15]. We performed a homology modelling of the missing loop structure with the Swiss-Model program using default parameters [16]. The insertion of the mutation and the analysis of the mutant were carried out using the UCSF Chimera program [17].

#### 2.3.2. Molecular Dynamic Simulations

For the BCKDK-wt and BCKDK-p.H162Q, respectively, we ran a 500 ns long all-atom MD simulation using the GROMACS program, version 2019 [18]. The Amber99SB forcefield was used to describe the protein. The parameter topology of the ligands was generated with the ACPYPE tool [19].

The systems (See Appendix A) were simulated in a cubic water box, and the minimum distance between any atom of the protein and the box wall was set up to 1.0 nm. The charges on the protein were neutralized by the addition of Na^+^ and Cl^−^ ions (0.145 M) to mimic physiological conditions, using the *genion* tool of the GROMACS package. The energy of the system was minimized using the steepest descent algorithm, performed for 50,000 steps with a maximum force constant value of 1000 kJ·mol^−1^ nm^−1^. The short-range electrostatic cut-off was set to 1 nm as well as the short-range van der Waals cut-off. Long-range electrostatic interactions were treated with the particle mesh Ewald (PME) method using a grid with a spacing of 0.16 nm. The cut-off radius for the Lennard–Jones interactions was set to 1 nm. NVT and NPT equilibrations for 100 ps each with a time step of 2 fs were performed. The systems were equilibrated at a temperature of 300 K and a pressure of 1 bar by a Berendsen thermostat [20] and Parrinello–Rahman barostat [21], respectively. Bonds involving hydrogen atoms were constrained using the LINCS algorithm [22]. The MD production was performed under the same conditions of the NPT ensemble with a time step of 2 fs for 500 ns. To gain a complete overview on the available structural data, we simulated 6 different systems using the same conditions but with different inhibitory ligands and structures (Appendix A). The conclusions of the present work apply to all the systems. For details on the analysis carried out on the 6 systems, see Appendix A.

## 3. Results

### 3.1. Genetic and In Silico Analysis

NGS WES analysis carried out on the patient identified the p.His162Gln variant in heterozygosis on the *BCKDK* gene. Sanger sequencing confirmed the same heterozygous substitution both in the patient and in his father.

This variant was never reported in the literature previously, nor in the population and clinical databases searched. In addition, the variant was predicted to be pathogenic by most prediction tools used, supporting a deleterious effect on the gene or gene product (SIFT: damaging, Polyphen2 HVAR: probably damaging, MutationTaster: damaging, MutationAssessor: predicted functional (high), FATHMM Pred^©^: tolerated, FATHMM MKL Coding Pred©: damaging). The variant was also predicted to be pathogenic by the BayesDel, addAF, DANN, DEOGEN2, EIGEN, LIST-S2, and M-CAP prediction tools.

### 3.2. Computational Analysis of the Mutation

As the 3D structure of BCKDK has been extensively characterized, we mapped His162 and its corresponding mutation on the structure of BCKDK (Figure 1). 

It can be appreciated that His162 residue is located in the binding cavity of the natural allosteric inhibitor of the protein. This region is important, because BCAA homeostasis is controlled by BCKDC, which is negatively regulated by BCKDK, which, on its turn, is inhibited by α-ketoisocaproate (KIC), resulting in the activation of BCKDC (Figure 2).

In this regard, Tso and collaborators showed that α-ketoisocaproate (KIC) binds to an allosteric binding site of BCKDK, leading to helix movements that, consequently, induce changes in the active site cleft and in the putative lipoyl-binding site (Figure 1) [15]. They also demonstrated, in vivo, that in the presence of KIC, the activity of BCKDC is increased, which leads to a reduction in the plasma concentration of BCAAs [15]. In fact, the inhibition of BDKDK by KIC reduces the inhibition of BDKDC, which then remains active. However, the molecular mechanism of the regulation of BCKDC by its physiological substrate (KIC) has, until now, never been clarified [15]. Therefore, we ran extensive atomistic molecular dynamics (MD) simulations on the wild-type (BCKDK) and mutated (p.His162Gln) BCKDK, with different inhibitors (Appendix A), with the aim of characterizing the molecular determinants underlying this mechanism of inhibition and how it changes in the presence of the p.His162Gln variant (Figure 3A). We first ran the simulations on the crystallographic inhibitors in order to validate our methodology. Indeed, the results of these simulations are in agreement with the results obtained by Tso and collaborators [15]. For more details regarding the MD simulations, see the SI. Our MD simulations revealed a particular behavior of the link between the α_4_ and α_5_ helices (Figure 1) that, in the presence of the inhibitors, induces the movement of the helices, as proposed by Tso et al., which consequently cause changes in the active site cleft and in the putative lipoyl-binding site. This agreement with the experimental results prompted us to run the simulations of the BCKDK-p.His162Gln mutant (SI). Here, instead, it can be observed that the amine group of the glutamine residue (Gln162) is able to interact with the oxygen of the backbone of an aspartate residue of the link between α_4_ and α_5_ helices (Asp164) (Figure 3A). The resulting bending angle between α_4_ and α_5_ helices leads to a conformation change (Figure 3B) that may contrast the effect of the inhibitor by inducing a movement in the opposite direction than in wild-type. Indeed, the presence of the mutation changes the dynamical properties of both helices (See SI). These results, within the limitations of the method, seem to point out that the effect of the inhibitor on BCKDK may be reduced by this mutation, resulting in its activation with subsequent inactivation of BCKDC and increase in plasmatic BCAAs, which is in agreement with the experimental data reported by Tso and collaborators [15]. For more details regarding the simulations, see Appendix A.

### 3.3. Follow-Up Data

During the clinical follow up, the patient presented a regular growth and an adequate psychomotor development, and he regularly underwent clinical examinations and neuropsychiatric evaluation, which yielded results in the normal range. The brain magnetic resonance imaging (MRI) performed at 2 years of age, as well as the electroencephalographic (EEG) evaluation, ruled out any CNS involvements. Additionally, the investigations performed in the patient’s father yielded normal results, including the brain MRI. An occasional stutter was observed in the patient, which was reported as a familiar finding in the paternal lineage and improved with logopedic intervention. During episodes of fever and intercurrent infections, the patient never developed metabolic ketoacidosis or neurological symptomatology, and the plasma values of BCAAs and alloisoleucine did not rise to significantly high values.

In view of the good clinical control and biochemical values and the normal neurological examination and cerebral MRI of the father, isoleucine and valine supplementations were gradually discontinued, as was thiamine supplementation, and a gradual increase in dietary protein intake was possible. From the age of 6 years, the protein intake was liberalized, while maintaining weekly and then bimonthly checks by dosing the amino acids leucine, isoleucine, valine, and alloIsoleucine on DBS at home (Figure 4).

## 4. Discussion

In the last few years, many kinases’ mutations, for tyrosine or serine, have been identified as key pathogenetic factors for many inherited disorders involving cell regulation (tumors) and metabolic and endocrinological processes. More than 500 distinct kinases are encoded by 2% of all human genes. Most of the kinase-related disorders involving the nervous and immune systems have an autosomal recessive pattern, while disorders involving skeletal, hematological, vascular, endocrinological, and metabolic processes show a dominant pattern. For instance, in metabolic kinasopathies, mutations in the insulin tyrosine kinase receptors lead to autosomal dominant insulin resistance with type 2 diabetes, metabolic dyslipidemia, lipodystrophy, and hepatic steatosis, whereas several loss-of-function mutations in fibroblast growth factor receptor 1 (also a tyrosine kinase) cause hypogonadotropic hypogonadism [23,24]. Lahiry et al. reviewed mutations in the tyrosine kinase family related to a wide range of phenotypes, whereby the severity of the disease phenotype may depend on the location of the mutation in the respective kinase gene. It is also possible that a mutation in different kinase genes leads to the same disease phenotype (locus heterogeneity), and from a functional point of view, kinase mutations can be classified as loss of function or gain of function [23]. In this regard, Zhang et al. reported two patients with hypokalemic salt-losing tubulopathies (SLTs) with inactivating mutations in the WNK1 kinase, although usually intronic heterozygous mutations on *WNK1* are known to cause pseudohypoaldosteronism type II, the opposite phenotype of SLTs [25].

Our study describes the identification of a heterozygous mutation in the gene *BCKDK* involved in the BCAA catabolism in a patient with biochemical characteristics compatible with mild MSUD, but with no mutations in the genes coding for the functionally catalytic subunits of BCKDC. 

BCKDK has been isolated and characterized, [26] and it is considered the key regulator of BCKDC activity [2,15]. BCKDK regulates the activity of the branched-chain amino acid catabolic pathway by phosphorylating, and thereby inactivating, BCKDH, the rate-limiting enzyme of the pathway, preventing the breakdown of BCAAs when these are necessary for protein synthesis. Therefore, modulation of BCKDK activity constitutes a major mechanism for BCAA homeostasis in vivo, offering a therapeutic target for ameliorating the accumulation of BCAAs in disease conditions [15,26].

BCKDK is allosterically regulated by BCKAs, in particular the KIC that inhibits the kinase promoting BCAA disposal when they are in excess and the conservation of these essential amino acids when they are less available. BCKDK also appears to be regulated by the association with BCKDC, although the molecular mechanism has not yet been established [27].

Novarino and colleagues tested, in 2012, a phosphor-specific antibody to the BCKDK phosphorylation site at residue 292 of E1α, supporting that patients with BCKDK mutations may lack basal, negative regulation of the BCKDH activity [28,29].

Recent studies show that loss-of-function mutations in *BCKDK* leading to excess rather than restricted BCAA oxidation may lead to autism spectrum disorder with epilepsy in humans [28]. The patients described by Novarino and colleagues’ studies showed notably lower levels of plasma BCAAs with respect to reference ranges. Another study in 2014 identified two unrelated children with two novel *BCKDK* mutations and a phenotype characterized by persistently reduced levels of BCAAs in body fluids and neurobehavioral abnormalities, developmental delay, and microcephaly, partially improved by a protein-rich diet plus oral BCAA supplementation [30]. Joshi and colleagues demonstrated, in 2006, that mice deficient in BCKDK display increased basal activity of BCKDC and consequently reduced levels of BCAAs in various tissues. These mice were healthy at birth but developed growth retardation and neurological abnormalities in adulthood, such as tremors, epileptic seizures, and autism phenotype. This phenotype in mice could be recovered by supplementation with BCAA-enriched diet [31]. The underlying molecular mechanism remains unknown, but it is supposed to be due to an alteration of several pathways, including the brain-expressed amino acid transporter network, underscoring the importance of BCAA homeostasis for normal brain functions [31]. These findings were confirmed by a spontaneous missense mutation in *BCKDK* found in a rat, affecting both the central and peripheral nervous systems [32]. With the exception of the *BCKDK* gene, MSUD-causing human mutations have been documented in each of the BCKDC components’ genes, as well as in the BCKDP gene [5,33]. Hence, to the best of our knowledge, this is the first time that BCKDK is called into question in a suspected MSUD phenotype.

In fact, we have identified in our patient a novel heterozygous p.His162Gln variant in the *BCKDK* gene, resulting in the amino acid substitution His to Gln at codon 162, homologous to His at codon 132 in rat BCKDK (rBCKDK) (Appendix A). The results of our computational analysis permit us to reasonably speculate that the effect of known inhibitors, such as KIC, α-chloroisocaproate, and phenylbutyrate, on BCKDK may be reduced by the His162Gln mutation, resulting in BCKDK activation. This amino acid substitution, although in heterozygosity, could therefore lead to a gain-of-function effect for the enzyme with the mutated allele acting in a dominant way by activating more BCKDK activity. This may lead, eventually, to excessive inhibition of BCKA dehydrogenase, resulting in increased plasma levels of BCAAs and likely in a MSUD phenotype, at least biochemically. Although our assumption is based on robust computational methods, further analysis of enzymatic assays of BCKDC activity in cultured fibroblasts will be necessary to confirm the hypothesis.

Meanwhile, in support of our study, even if in a different regulatory way, recent studies found a mutation on the mitochondrial protein phosphatase 2Cm (PP2Cm) in a patient with a mild-variant MSUD phenotype and with no nucleotide changes identified in any of the normally affected genes [34]. PP2Cm is encoded by the *PPM1K* (NM_152542.5) gene and is apparently a critical regulator of BCAA catabolism in mice and humans through the dephosphorylation of BCKDH [35,36,37,38]. A study by Oyarzabal et al. confirms that defective activation of BCKDC via a defect in PP2Cm production leads to a significant increase in plasma concentrations of BCKAs and BCAAs, reaching levels associated with a mild form of MSUD [34]. Alterations on regulatory proteins, and in this case a loss of function in phosphates and a gain of function in kinase, even if acting in apparent opposite ways, may cause similar pictures pointing out the possible pathogenetic role of such mutations [23].

Going back to our family case study, we identified through NBS a patient with a heterozygous mutation in the *BCKDK* gene inherited from his father. Both have a similar biochemical phenotype characterized by slightly elevated plasma BCAAs and Alloile, the pathognomonic marker for MSUD diagnosis, confirming the segregation of genotypes and phenotypes and showing a dominant pattern. Looking into other family members and, in particular, the biochemical phenotype of the paternal grandfather, we ruled out the presence of a significant alteration in BCAA metabolism, through a protein loading test, which showed plasma values of BCAAs and Alloile within the reference range. In the end, it should be considered that blood BCAA levels may be regulated by multiple factors influencing degradation pathways, such as obesity, insulin resistance, and aging and may change with protein intake [39].

## 5. Conclusions

Our study is the first to propose a regulatory inactivation of BCKDC due to a dominant gain-of-function mutation in the human *BCKDK* gene, a candidate gene previously never found to be involved in MSUD pathogenesis. Based on literature cases, our patient was considered to be affected by a mild variant of MSUD and therefore at possible risk of metabolic decompensation during stressful situations. Nevertheless, based on the course of his disease and the natural history of the father, a benign biochemical variant cannot be excluded. Further studies, such as enzymatic assays, and a longer follow-up are needed to elucidate whether the phenotype caused by this variant has clinical relevance or only biochemical.

Awareness of this regulatory defect of BCKDC is very important in the genetic diagnosis of MSUD patients, especially for children with branched-chain amino acid alterations during newborn screening without a definitive genetic diagnosis.

## Figures and Tables

**Figure 1 genes-13-00233-f001:**
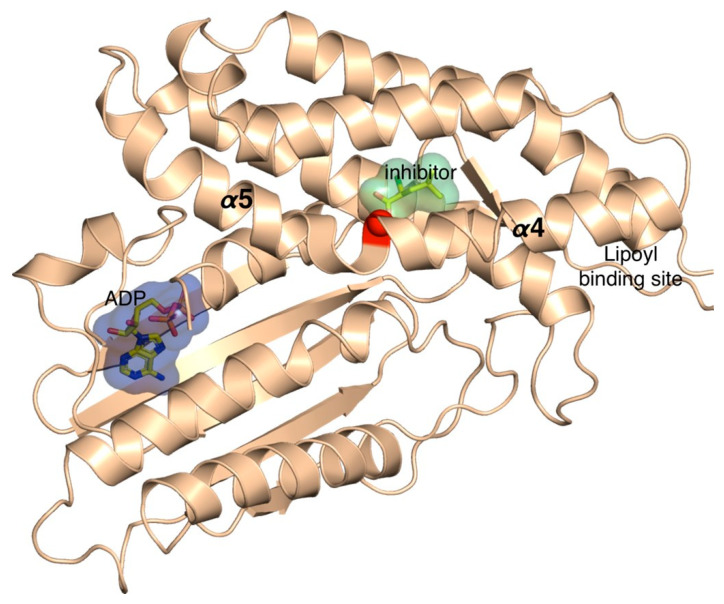
BCKDK three-dimensional structure cocrystallized with ADP and the synthetic inhibitor S-α-chloroisocaproate (PDB accession code: 3TZ4). The red ball represents the location of the mutation p.His192Gln.

**Figure 2 genes-13-00233-f002:**
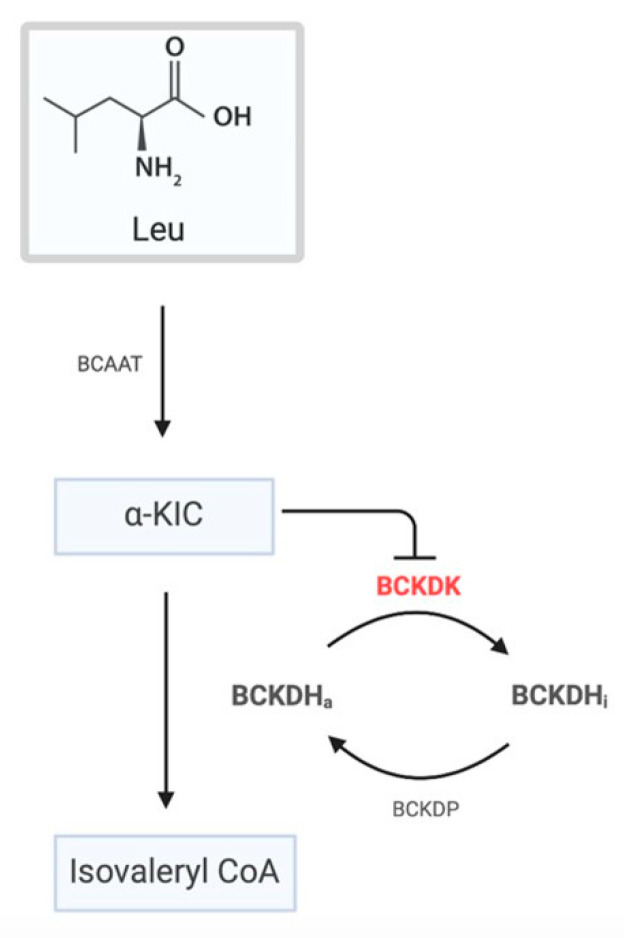
Schematic representation of the regulation of BCKDC. BCAAs undergo a transamination reaction by BCAAT. The resulting α-ketoacids are then oxidatively decarboxylated by BCKDH. The overall activity of BCKDH is regulated by phosphorylation/dephosphorylation by BCKDK and DCKDP, respectively. On its turn, BDKDK is negatively regulated by the α-ketoacids, leading to an increase in the activation of BCKDH (dephosphorylated state).

**Figure 3 genes-13-00233-f003:**
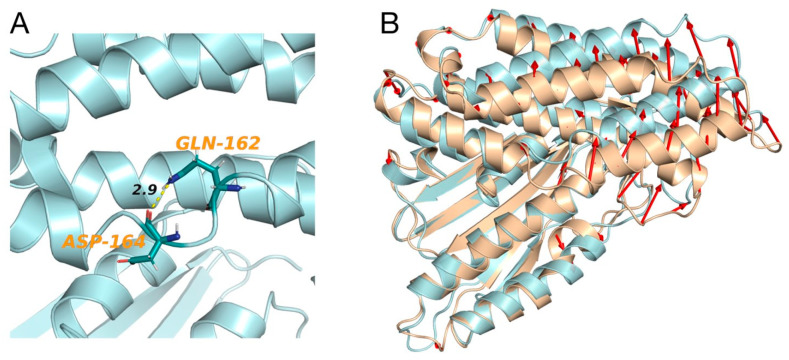
**(A**) Interaction (distance in angstrom) between Gln162 and Asp164 of BCKDK-p.His162Gln after 500 ns of MD simulation. (**B**) Superimposition of BCKDK-wt (light brown) and BCKDK-p.His162Gln (light blue) after MD simulations. The red arrows represent a deviation higher than 3 Å.

**Figure 4 genes-13-00233-f004:**
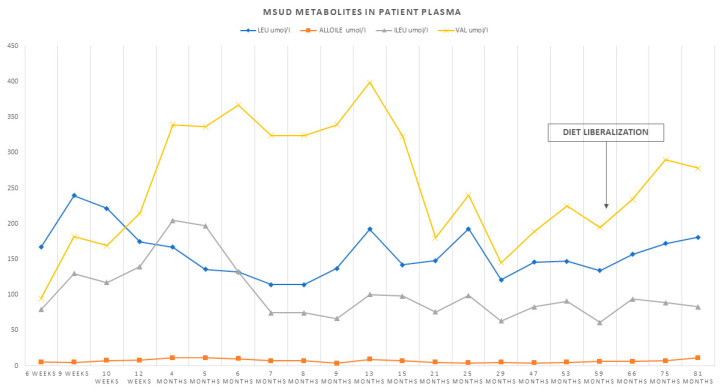
Graphical representation of the trend of BCAAs and alloisoleucine during follow-up and after protein intake liberalization.

**Table 1 genes-13-00233-t001:** DBS and plasma BCAAs and alloIsoleucine of the patient at diagnosis and follow-up.

BCAAs	LEU	ILE	VAL	ALLOILE
Neonatal DBS (reference range values)	–	–	<250 µmol/L	<2.0 µmol/L
	–	–	214	8.2
DBS (reference range values) at follow-up *	<200 µmol/L	<100 µmol/L	70–267 µmol/L	<2.0 µmol/L
	44–324	4–157	62–514	0–14
Plasma (reference range values) at diagnosis	77–195 µmol/L	38–99 µmol/L	130–335 µmol/L	<2.0 µmol/L
At diagnosis	428	207	418	17.0
Plasma (reference range values) at follow-up	75–127 µmol/L	39–65 µmol/L	158–291 µmol/L	<2.0 µmol/L
At follow-up	121–193	63–99	145–323	4–11

BCAAs and alloIsoleucine of the patient at neonatal DBS and plasma confirmation and during DBS and plasma follow-up, specifying the min–max values found and our laboratory reference values. * DBS at follow-up was used for domiciliary monitoring. Abbreviations: DBS, dried blood spot; BCAAs, branched-chain amino acids; LEU, leucine; ILE, isoleucine; VAL, valine; ALLOILE, alloisoleucine.

**Table 2 genes-13-00233-t002:** Plasma BCAAs and alloIsoleucine of relatives.

Plasma BCAAs (Reference Range Values)	LEU (77–195 μmol/L)	ILE (38–99 μmol/L)	VAL (130–335 μmol/L)	ALLOILE (<2.0 μmol/L)
Patient	**428**	**207**	**418**	**17.0**
Father	**384**	**156**	**403**	**14.0**
Paternal grandfather	**234**	**111**	315	**3.6**

Abbreviations: BCAAs, branched-chain amino acids; LEU, leucine; ILE, isoleucine; VAL, valine; ALLOILE, alloisoleucine. In bold pathological values.

**Table 3 genes-13-00233-t003:** Plasma BCAAs and alloIsoleucine after protein load test.

Plasma BCAAs (Reference Range Values)	LEU (77–195 μmol/L)	ILE (38–99 μmol/L)	VAL (130–335 μmol/L)	ALLOILE (<2 μmol/L)
Paternal grandfather	109	74	193	1
Father	**242**	**181**	**425**	**14**

Abbreviations: BCAAs, branched-chain amino acids; LEU, leucine; ILE, isoleucine; VAL, valine; ALLOILE, alloisoleucine. In bold pathological values.

## Data Availability

Data are contained within the article or supplementary material.

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
