# Peer review of "A Gain-of-Function Mutation on BCKDK Gene and Its Possible Pathogenic Role in Branched-Chain Amino Acid Metabolism"

_genes, 2022, doi:10.3390/genes13020233_

Round 1

Reviewer 1 Report

In this study, A. Maguolo and colleagues described a rare variant in a conserved region of BCKDK (His162Gln), potentially a gain-of function mutation responsible of false positive newborn screening and altered levels of alloisoleucine (key biomarker of MSUD), along with slightly elevated branched-chain amino acids levels. This is very interesting at least in a biochemical point of view but requires more experimental data to confirm the computational model.

Comment 1: Even if the computational analysis is promising, a BCKDC activity in cultured fibroblasts (in the presence or not of natural or synthetic inhibitors of BCKDK such as alpha-chloro-isocaproate or KIC or PB) is lacking to support your in-silico hypothesis (as suggested by yourself, see line 97 and conclusion).

A functional assay is required to support the hypothesis, especially because you are describing a single family case.

Comment 2: Regarding Table 1 : Could you show more concentration values for the 4 amino acids, especially since you mention that you measured it during the follow-up. It would be informative to show the concentrations evolution between the NBS values over the time of follow-up. I’m surprised to see that AlloIle in the grandfather sample showed in table 1 (3.6) is higher than after the protein loading test (1) in table 2/II.

Comment 3: Line 114: “Uninformative urinary organic acid” is described.  Could you please give more information about the measured concentrations of branched-chain hydroxy- and keto-acids? (including reference values) Did you also perform urine organic acids analysis following the protein load test? Did you assess branched-chain amino acids in urine samples as well?

Comment 4: Did you exclude the presence of mutations in BCAT2 and PPM1K? Two known conditions for mild MSUD-like amino acid profile.

Comment 5: Lines 320-327, PPM1K deficiency is indeed a cause of mild MSUD profile. However, does this condition sustain your hypothesis?

Comment 6: I would suggest to reformulate and shorten a bit the title

Reviewer 2 Report

In the manuscript “A gain of function mutation on BCKDK gene: from new-born screening new insights in regulation of branched-chain amino acid metabolism” the authors identified the p. His162Gln variant of the BCKDK gene in a new-born with a biochemical phenotype of a mild form of Maple Syrup Urine Disease (MSUD). The variant was predicted as pathogenic by most prediction tools used.  Furthermore, by computational and simulation analyses the authors sustain that the effect of known inhibitors of the BCKDK protein may be reduced by the His162Gln mutation, resulting in its increased activity. This would determine a subsequent inactivation of BCKDC and an increase of plasmatic branched-chain amino acids levels. The authors propose that the heterozygous His162Gln mutation could exert a gain-of-function effect, and BCKDK gene could be referred to as a new gene responsible for MSUD.

Main Criticism

The hypothesis advanced by the authors on the gain-of function effect in the protein with the identified mutation should be better supported by literature data. The instances reported in the discussion section concern the effect of mutations of the BCKDK gene in homozygosity that abolish enzymatic activity, a situation completely different from the described His162Gln mutation found in heterozygosity. Furthermore, the references report mutations in BCKDK that are associated with phenotypes very dissimilar from MSUD, since they are strictly related with neurological alterations and determine reduced levels of BCAAs in the patients.

Therefore, the addition of specific references on mutations in other kinases, in which the different mutations can either result in gain-of-function or loss of function effect, could better support the validity of the authors' hypothesis.

Round 2

Reviewer 1 Report

The authors have answered my comments well. I have no additional comment.